# Accuracy Assessment of Geometric-Distortion Identification Methods for Sentinel-1 Synthetic Aperture Radar Imagery in Highland Mountainous Regions

**DOI:** 10.3390/s24092834

**Published:** 2024-04-29

**Authors:** Chao Shi, Xiaoqing Zuo, Jianming Zhang, Daming Zhu, Yongfa Li, Jinwei Bu

**Affiliations:** Faculty of Land Resources Engineering, Kunming University of Science and Technology, Kunming 650093, China; 20222201127@stu.kust.edu.cn (C.S.); kmustzjm@kust.edu.cn (J.Z.); 11301066@kust.edu.cn (D.Z.); yfli@kust.edu.cn (Y.L.); b_jinwei@kust.edu.cn (J.B.)

**Keywords:** synthetic aperture radar, geometric-distortion identification, layover and shadow map, R-Index, pixel-neighbor gradient, highland mountainous regions

## Abstract

SAR imagery plays a crucial role in geological and environmental monitoring, particularly in highland mountainous regions. However, inherent geometric distortions in SAR images often undermine the precision of remote sensing analyses. Accurately identifying and classifying these distortions is key to analyzing their origins and enhancing the quality and accuracy of monitoring efforts. While the layover and shadow map (LSM) approach is commonly utilized to identify distortions, it falls short in classifying subtle ones. This study introduces a novel LSM ground-range slope (LG) method, tailored for the refined identification of minor distortions to augment the LSM approach. We implemented the LG method on Sentinel-1 SAR imagery from the tri-junction area where the Xiaojiang, Pudu, and Jinsha rivers converge at the Yunnan-Sichuan border. By comparing effective monitoring-point densities, we evaluated and validated traditional methods—LSM, R-Index, and P-NG—against the LG method. The LG method demonstrates superior performance in discriminating subtle distortions within complex terrains through its secondary classification process, which allows for precise and comprehensive recognition of geometric distortions. Furthermore, our research examines the impact of varying slope parameters during the classification process on the accuracy of distortion identification. This study addresses significant gaps in recognizing geometric distortions and lays a foundation for more precise SAR imagery analysis in complex geographic settings.

## 1. Introduction

Interferometric Synthetic Aperture Radar (InSAR) technology, renowned for its wide application spectrum, high precision, and all-weather capabilities, is instrumental in the rapid response to sudden geological events such as earthquakes and volcanoes [1,2], and is indispensable for long-term monitoring of crustal movements, glaciers, landslides, urban and mining subsidence, and the stability of artificial structures [3,4,5,6,7,8]. This technique is exceptionally valuable in disaster prevention, geological surveys, engineering construction, and precision mapping. However, the side-looking imaging geometry of radar satellites can introduce geometric distortions into SAR imagery [9,10], which affects the accuracy of the data.

Geometric distortions in SAR imagery can be categorized into four main types based on the relative geometry between the satellite and the terrain: foreshortening, layover, enhanced resolution, and shadow [11]. Areas of enhanced resolution improve terrain feature differentiation; foreshortened areas suffer from resolution degradation that reduces the quality of deformation monitoring outcomes; and the superposition of signals in layover areas coupled with signal loss in shadow areas leads to phase discontinuities, impacting phase unwrapping errors [11,12]. SAR imagery is particularly susceptible to geometric distortions in high mountains and deep valleys. Accurate detection of these distortions is crucial for acquiring high-quality SAR images [13], analyzing terrestrial backscatter characteristics, and enhancing the accuracy of InSAR deformation monitoring [14]. It also aids in developing more efficient InSAR deformation monitoring strategies, thus saving time and costs [15].

Current methods for classifying geometric distortion types in SAR imagery are primarily divided into two categories [16,17]: those based on the statistical characteristics, and those utilizing the relative geometric relationship between the sensor and the Earth’s surface for identifying and extracting distorted areas. Results based on statistical characteristics are heavily influenced by the surface features of the study area, which limits their universality. Kropatsch and Strobl explored the origins of SAR geometric distortions from an imaging principle perspective in 1990, proposing a method that employs the Digital Elevation Model (DEM) to generate SAR layover and shadow maps [18]. Notti et al., considering SAR imaging geometry and topographical influences, introduced the R-Index model for geometric distortion identification [19,20,21], which predicts the distribution of potential permanent scatterers (PS). Chen et al. refined the LSM with a DEM-based neighborhood-gradient algorithm, developing the P-GN model [22]. This LSM method is highly regarded for its accuracy, interpretability, and practicality, as it anticipates layover and shadow zones using parameters such as satellite position, flight attitude, and ground elevation [23]. The R-Index method identifies geometric distortion areas by computing the interrelation between slope, aspect, and satellite parameters [24,25,26]. Based on satellite and terrestrial geometry, the P-NG technique discerns geometric distortions by assessing the elevation gradient change in adjacent pixels, accurately differentiating between active and passive regions while accommodating the complexity of geometric distortion overlaps. However, the complex P-NG algorithm is computationally inefficient and unsuitable for extensive study areas. Existing distortion-identification methods lack sufficient refinement in distortion classification, particularly in the detailed delineation of subtle distortions; additionally, few studies have thoroughly investigated the precision of such identifications and their impact on monitoring results from a foundational perspective [27,28,29].

Addressing the limitations of conventional methods, this paper presents a novel identification strategy termed LG, which specifically targets subtle distortions and achieves a more refined classification by analyzing local incidence angles (LIAs) alongside ground-range slope. This research was conducted in the challenging high-altitude convergence zone of the Xiaojiang, Pudu, and Jinsha Rivers, where significant topographical variations and frequent geological hazards create an ideal setting for testing and comparing distortion identification methods.

The paper details four techniques—LSM, the R-Index, P-NG, and the newly introduced LG approach—which are applied to identify and delineate geometric distortions in SAR imagery within the selected study area. Through empirical analysis, the performance of these methods is evaluated at various distortion levels, with effective monitoring-point density serving as the benchmark for performance assessment. Additionally, the theoretical foundations of each method are explored, emphasizing their practical implications for real-world SAR image-distortion identification. The experimental results aim to provide methodological insights for enhancing SAR image processing and furthering the application of these technologies in mountainous terrains. As the study progresses into comparative analyses, it sheds light on the intricate roles played by slope, ground-range slope, and the satellite’s LIAs in determining the accuracy of geometric distortion identification. The findings offer actionable insights into the adaptability of each method under diverse environmental conditions and propose directions for future research aimed at increasing precision in InSAR applications.

## 2. Materials and Methods

### 2.1. Materials

#### 2.1.1. Study Area

The research area extends from the northeastern part of Kunming City to the northwestern edge of Huize County, encompassing the basins of the Xiaojiang, Pudu, and Jinsha rivers over an approximate area of 6151 square kilometers (as illustrated in Figure 1). The region is characterized by complex terrain, with 25% of the area exhibiting between 20° and 30°, and 39.9% featuring slopes greater than 30°. About 67.24% of the surface elevations within the study area range between 2000 and 3000 m, with the maximum elevation difference exceeding 3600 m. Vegetation cover is sparse, with 38.02% of the area containing tall, densely canopied vegetation [30]. This area is influenced by the Xiaojiang and Pudu River fault zones [31,32], and includes Quaternary rift basins among other geological features, which results in a region with active geological structures, fractured rock formations, and poor stability, making it prone to frequent geological disasters [33]. Within this context, Dongchuan alone has documented 491 geological disaster points, including 280 landslides, 25 collapses, 172 debris flows, and 14 ground subsidences, and features a debris-flow gully distribution density of 43.7 strips per 1000 square kilometers.

For this study, four representative local areas within the research region were selected to better investigate the coupling between topographic factors, distortions, and InSAR monitoring results, as depicted in Figure 1c. These areas are described as follows: A1 is an east-facing slope devoid of tall trees; A2 is a valley characterized by sparse vegetation; A3 is a northwest-southeast oriented ridge covered with dense vegetation; and A4 is a north-south oriented gully terrain with sparse vegetation.

#### 2.1.2. Data

Sentinel-1 operates in a polar orbit and is equipped with sophisticated C-band Synthetic Aperture Radar (SAR) instrumentation. This setup enables all-weather, day-and-night imaging capabilities, which are crucial for monitoring the Earth’s surface. Currently, Sentinel-1 is extensively used for continuous surveillance of regional ground deformations across large areas, aiding in the identification of zones susceptible to geological hazards [34].

Satellite assessments play a critical role in analyzing various disaster scenarios, evaluating their impacts and devising targeted mitigation strategies based on robust terrestrial data. In this study, SAR data from both ascending and descending tracks of the Sentinel-1 satellite were utilized to monitor surface deformation. The key parameters of the Sentinel data used in this study are presented in Table 1, and the spatial extent of the imagery is depicted in Figure 1b.

The study incorporated the Generic Atmospheric Correction Online Service for InSAR (GACOS) to correct atmospheric delays in the InSAR processing workflow. GACOS utilizes an iterative tropospheric decomposition model to distinguish between stratified and turbulent signals within the total delay, thereby facilitating the generation of Zenith Total Delay (ZTD) maps to correct atmospheric errors observed in InSAR monitoring [35,36,37].

Furthermore, this research employed a 5 m resolution, resampled DEM to effectively eliminate topographic phase errors. This DEM was also instrumental in identifying geometric distortions within the study area.

To detect large-scale geometric distortions, the local incidence angle (LIA) for each surface element was extracted from Sentinel-1 SLC data. The variations in LIA are systematically presented in Figure 1d for the ascending track and Figure 1e for the descending track, with the ascending LIA ranging from 41.72° to 45.96°, showing a consistent increase from west to east, and the descending LIA ranging between 33.94° and 39.69°.

### 2.2. Methods

#### 2.2.1. SBAS-InSAR 

The Small Baseline Subset InSAR technique (SBAS-InSAR), pioneered by Berardino in 2002 [38,39], relies on the principle of utilizing short baseline lengths to assemble a large array of SAR data into subsets that include multiple master images. This configuration keeps the baseline length of each interferogram within a subset below a critical threshold, and maintains short temporal baselines, while allowing for greater separation between SAR images across different subsets. This strategy effectively addresses the decorrelation issues caused by temporal and spatial discrepancies, making SBAS-InSAR particularly well-suited for monitoring surface deformation in distributed scatterer environments, such as mountainous terrain [40,41].

The study area, situated in a high-altitude mountainous region, presents significant topographical variations and severe topographic errors. Additionally, interferogram quality is often compromised by snow cover in high mountain areas. The SBAS method mitigates these effects by utilizing small-baseline interferometric combinations that do not require the selection of persistent scatterers with high coherence. This allows for a more continuous and comprehensive deformation field, making SBAS-InSAR the preferred technique for acquiring high-quality deformation time series for spatial and temporal monitoring in this mountainous context.

In SBAS-InSAR, a single image is designated as the common master image. Subsequent images that meet predefined temporal- and spatial-baseline thresholds are used to form a time series of differential interferograms. By selecting high-coherence interferometric targets and applying the least squares method, the deformation series for each small subset is extracted, including phases for deformation, atmospheric delay, and orbital errors. Due to the multiple master images used in SBAS-InSAR, a rank deficiency may occur in the system of equations during the joint inversion of the subsets. To resolve this, singular value decomposition is applied to solve jointly the following: multiple small-baseline sets, compute phase residuals, and perform residual separation. This process ultimately yields the time series of surface deformation in the radar line-of-sight direction, as depicted in Figure 2. The SBAS processing in this study is conducted using SARscape version 5.6.

#### 2.2.2. SAR Geometric-Distortion Identification Method

As outlined in Table 2, this method classifies distortions based on the relationship between ground-range slope and local incidence angle, identifying layover and shadowing as severe types of distortion [10,42,43,44]. The conversion of terrain slope into ground-range slope is mathematically expressed in Equation (1), where S represents the slope, *α* is the satellite azimuth angle, and A is the slope aspect.
(1)X=|S∗sin(α−A)|

As illustrated in Figure 3, within the foreshortening zone, the angle between the radar line-of-sight and the surface normal decreases, which alters the pixel ground resolution perpendicular to the satellite flight direction, resulting in foreshortening. In steep fore slopes, radar signals reaching the summit travel a shorter path than those directed towards the base, causing the summit’s echo to overlap prematurely with the base. This phenomenon, known as layover, predominantly affects areas directly in front of the base, behind the peak, and is passively influenced by their distance from the satellite. Shadow occurs on steep leeward slopes where pixels with a slope angle greater than or equal to the complement of the local incidence angle are in active shadow, rendering them undetectable due to a lack of signal reception, with signal levels in processed images comparable to system thermal noise. In certain conditions, less-steep slopes than the complement of the local incidence angle may exhibit ‘enhanced’ imaging, where backslope resolution can exceed that of flat terrain. Equation (1) suggests that resolution is optimal when the radar beam skims closely over the terrain.

Compared to imagery captured by optical sensors, SAR imagery involves a more complex mapping geometry. The LSM approach considers terrain elevation, slant range, and slope variations along the ground range. It considers the satellite’s position and the interaction between different types of distortions, categorizing them into seven classes: valid data (subtle distortion), near-passive layover, far-passive layover, active layover, layover shadow, active shadow, and passive shadow.

As depicted by Equation (2), the R-Index model integrates the DEM with the satellite’s local-incidence and flight-azimuth angles to calculate the R index for each ground pixel. This model quantifies the impact of terrain on SAR imagery.
(2)R=cos(θ+S∗sinAα)

In this formula, R represents the R index where $R\in \left(\cos\left(\theta +90{}^{\circ}\right)\right.\left.,1\right]$, S denotes the terrain slope, Aα adjusts for the slope orientation relative to the satellite’s flight azimuth, and *θ* is the satellite’s LIA. An R value less than 0 defines the pixel as being subjected to active layover. When the slope and the LIA value are complementary angles, the boundary condition is denoted as $R_{\theta }$, which delineates the transition between foreshortening and good visibility. Pixels with R values ranging from 0 to $R_{\theta }$ exhibit foreshortening, while those with R values greater than $R_{\theta }$ indicate good visibility.

Building on Colesanti’s proposed approach, the pixel-neighbor gradient (P-NG) method, described by Equation (3), replaces the ground-range slope with the slope to consider the interaction between adjacent terrain features. This method characterizes the region of good visibility within the R index as resolution enhancing and introduces techniques to identify near-passive layover, far-passive layover, and passive shadow. Furthermore, pixels where the slope equals the complement of the local incidence angle are classified as having the worst resolution.
(3)$$NG=\frac{\left|h_{ij}-h_{ik}\right|}{gr\cdot \left|j-k\right|}$$
$\left|h_{ij}-h_{ik}\right|$ represents the elevation difference between point $\left(i,j\right)$ and $\left(i,k\right)$ within the area characterized by non-active, subtle distortion, while $gr\cdot \left|j-k\right|$ denotes the ground distance between these points, with gr being the planimetric resolution of the DEM. The NG is compared with the tangent of the slope $S_{ij}$, which marks the boundary point for severe active distortion at coordinates $\left(i,k\right)$. In this study, following the P-NG method, slope is used rather than ground-range slope for distortion identification. The implications of using slope versus ground-range slope in distortion delineation will be discussed further in the Discussion section. If $NG>\tan S_{ij}$, then the point $\left(i,k\right)$ is classified as passive layover; conversely, if $1/NG<\tan S_{ij}$, it is classified as passive shadow.

Considering the strong interpretability and repeatability of the LSM algorithm and its use in classifying distortions based on slope and ground-range slope, this study integrates these two approaches to comprehensively identify geometric distortions. We propose a combined detection method, termed LG, which employs the following identification process:Utilize the LSM algorithm to classify six severe distortion types in pixels: active layover, near-passive layover, far-passive layover, layover shadow, active shadow, and passive shadow.For pixels with subtle distortions not categorized by LSM, employ geometric distortion principles to classify them as either foreshortening or resolution enhancing. This process is depicted in Figure 4.

#### 2.2.3. Validation of Identification Accuracy

The delineation of geometric distortions is crucial for quality assessment in deformation monitoring. To validate the accuracy of the distortion classification, this study evaluates this not only based on the distribution of distortion morphologies, but also corroborates these evaluations with deformation monitoring results. We define “effective monitoring point density” (EV) as the ratio of the number of deformation pixels (ND) within each distortion category to the total number of pixels (T) in that category. The specific formula is given as the following:(4)$$EV_{i}=\frac{ND_{i}}{T_{i}}\times 100\%$$
Here, *i* represents various types of geometric distortions. This point density metric illustrates the relationship between different types of distortions and the performance of InSAR deformation extraction, providing an intuitive display of classification results and aiding in the understanding of the accuracy of different geometric-distortion classification methods. Higher values of EV are typically observed in areas of foreshortening and resolution enhancing, where a larger number of deformation pixels are detected. Conversely, areas affected by shadow often yield lower EV values due to signal loss, resulting in fewer deformation pixels detected. In regions characterized by layover, the mixed signals contribute to a decreased effectiveness value. By calculating the proportion of deformation pixels within specific types of geometric distortions, we discern the relationship between various geometric distortions and the efficacy of deformation extraction.

## 3. Results

### 3.1. InSAR Processing

This study employed SBAS-InSAR technology to meticulously process Sentinel-1 satellite data spanning from January 2021 to December 2022. We integrated 60 ascending and 56 descending datasets to construct a comprehensive deformation velocity dataset representative of the radar line of sight (LOS), as depicted in Figure 5a,b. Here, negative values signify displacement moving away from the satellite sensor, while positive values indicate deformation moving toward the sensor.

Initial analyses focused on the annual average deformation-velocity maps, which delineated the average movement trends of the study area over the entire period. Distributed deformation velocities varied, with ascending data showing rates from −143 mm/year to 174 mm/year, and descending data from −186 mm/year to 125 mm/year. An analysis of the statistical distribution of deformation rates revealed that approximately 95.80% of the deformation in ascending data was concentrated within a rate range of ±20 mm/year, while about 84.83% of deformation in descending data fell within the same range. This distribution pattern suggests that minor deformations predominantly characterize surface activities, though regional deformation events are also evident. Correlations between deformation and land use types (shown in Figure 5c) indicated that SBAS-InSAR techniques struggled to capture ground deformation in areas with dense vegetation, glaciers, snow, and ground ice, due to significant decorrelation.

Further analysis as illustrated in Figure 6 identifies multiple deformation zones in newly developed areas at the northern foothills of Huize’s urban district. Subsidence in the central new residential area, labeled H1, is particularly severe, with an annual deformation rate nearing 70 mm/year. Expansion in the eastern industrial park, marked H2, and construction in the southern new residential zone, labeled H3, have also induced notable deformation. No significant deformation signals were observed in the old city district. Additionally, regular phase fringe changes were observed in the agricultural land in the central part of the county (blue framed area in Figure 6a), consistent with the distribution of plastic greenhouses. In the Jiangjia Gully area, significant deformation was observed in the historic landslide region. Ascending-orbit data recorded larger deformation signals due to the observation angle being parallel to the landslide plane, while descending orbit data presented a more concentrated deformation area perpendicular to the landslide surface. Meanwhile, northwest-facing slopes shielded by plateaus did not exhibit monitoring data.

Figure 7 depicts the deformation rate maps for ascending and descending orbits in the four local areas. Data from the ascending orbit for area A1 indicate numerous positive monitoring pixels, suggesting upward ground movement toward the southwest. Conversely, data from the descending orbit show monitoring pixels mainly distributed on the southeast-facing slope, indicating movement away from the satellite towards the west. Integrating these images from both trajectories reveals subsidence zones in the A1 area, with the massif exhibiting a movement trend towards the base of the slope. The monitoring data for area A2 are concentrated on the northwest-facing, flatter part of the mountain, with both ascending- and descending-orbit monitoring indicating movement towards the satellite, suggesting overall peeling of the mountain surface layer and local subsidence in the midsection. Moreover, the southwestern region displays ground uplift. Monitoring data for area A3 are relatively sparse, but notably, the pixels recording deformation values between −60 mm/year and −95 mm/year in descending orbit images are located within this zone. Area A4 exhibits a significant complementary phenomenon across ascending- and descending-orbit images. Monitoring pixels from ascending-orbit images are primarily found in areas of gentle relief and on the east-facing mountainside, whereas those from descending-orbit images are mainly on the western mountainside. Pixels from overlapping parts in both show positive and negative numerical differences, signifying a tendency for the east-facing slope to move toward the valley floor. 

### 3.2. Identification of Geometric Distortions

#### 3.2.1. Ascending-Track Identification Results

Figure 8a–d showcase the results of geometric distortion identification in Sentinel-1 ascending imagery using LSM, R-index, P-NG, and LG methods, respectively. Each method exhibits distinctive distribution characteristics, with layover distortion predominantly appearing on west-facing slopes and extensive layover/shadow detected across the western and central parts of the study area.

The LSM method successfully identified several typical distortion types including near-passive layover, active layover, far-passive layover, layover shadow, active shadow, passive shadow, and valid data. The spatial arrangement of these distortions demonstrates a regular pattern with clearly defined boundaries, particularly evident in areas with significant topographical variations, such as along the west bank of the Pudu River and on ridges and slopes near the Jinsha River.

The R-Index method simplifies distortion identification into three main categories: active layover, good visibility, and foreshortening. The active layover identified by the R-Index closely aligns with results recognized by the LSM, predominantly on slopes facing the satellite. This method provides an index of the geomorphological relationship between the satellite and the terrain, exploring the extent of distortions.

The P-NG method, on the other hand, does not consider layover shadow and categorizes subtle distortions as either foreshortening or resolution enhancing. This approach detects a broader range of severe distortions, extending beyond slopes perpendicular to the LOS, and identifies more distortions in areas of greater topographical relief. However, P-NG sometimes merges different types of geometric distortions, with some regions showing an overlap of active and passive shadow and others displaying only passive distortion without the corresponding active distortion.

The LG method aligns closely with the LSM regarding the distribution of layover and shadow regions, presenting layover predominantly on west-facing slopes (towards the satellite) and shadow on east-facing slopes (away from the satellite). The areas of foreshortening and resolution enhancing identified by the LG method correspond with those detected by the R-index, and, compared to P-NG, LG identifies fewer severe distortions.

#### 3.2.2. Descending-Track Identification Results

Figure 9a–d illustrate the geometric distortion outcomes identified by LSM, R-index, P-NG, and LG methodologies within the Sentinel-1 descending imagery. Due to shifts in the satellite’s viewing angle, distortions are more pronounced on east-facing slopes oriented toward the sensor. The LSM method for descending orbits detected a higher incidence of geometric distortions, particularly on the northeastern, eastern, and southeastern slopes facing the satellite, with a widespread distribution of layover especially on the western bank of the Pudu River and interwoven with less dense shadow on the eastern bank.

The R-index results showed fewer pixels of good visibility and a significant increase in active layover pixels compared to ascending orbits, primarily occurring along riversides and ridgelines. P-NG identified a broad range of severe distortions, notably along the banks of the Pudu and Xiaojiang Rivers and the southern shore of the Jinsha River, with extensive shadow distortion engulfing the eastern bank. The LG method’s descending-orbit results aligned with those of LSM in terms of severe distortion distribution, but it identified fewer severe distortions compared to P-NG, reflecting a more conservative approach similar to the R-index results. 

This analysis across Figure 8 and Figure 9 shows that each method consistently identified terrain-induced distortions, with variances in the degree and granularity of distortion detected. LSM delineated clear types of distortions, R-index excelled in identifying active layover, P-NG captured extensive severe distortions, and LG aligned with LSM and R-index in some aspects but was more conservative in identifying severe distortions, recalculating for slopes facing the satellite’s flight direction. 

#### 3.2.3. Quantitative Comparison of Identification Results

The statistical analysis of geometric distortion areas within the study region identified by different detection methods is presented in Table 3. Notable differences are observed in the proportions of two severe distortion types—layover and shadow—across the four identification methods. Active layover is the most prevalent severe distortion type in LSM, accounting for 3.36% in ascending tracks and 7.32% in descending tracks. The proportion of passive distortions is generally smaller than their active counterparts. The term ‘valid data’ encompasses both foreshortening and good visibility, representing images of SAR data with non-severe distortions as a unified category.

The P-NG method, by comparing LIA with slope rather than using the reduced ground-range slope, classifies more severe distortions on steep slopes. In ascending data, the proportion of near-passive layover is 3.3%, surpassing that of active layover, and the proportion of passive shadow also exceeds that of active shadow. In descending data, passive shadow accounts for 8.65% while active shadow accounts for only 1.15%. Conversely, the R-index, which only recognizes active layover and non-severe distortions, identifies the lowest proportion of active layover among all methods. The proportion of severe distortions identified by the LG method aligns with that of the LSM method, and the combined proportions of foreshortening and good visibility identified are equivalent to the proportion of the valid data area in the LSM method.

Terrain directly influences geometric distortions in SAR imagery, with changes in slope aspect and slope degree being critical factors. This study has compiled statistics on the distribution of various distortions across different terrain regions, as detailed in Table 4 (Slope) and Table 5 (Aspect).

In ascending imagery, 90.07% of the severe distortions identified by the LSM/LG method occurred on slopes ranging from 20° to 60°. The R-Index showed that active layover was predominantly located on slopes exceeding 40°, accounting for 96.74% of these instances. In contrast, the P-NG method reported that 93.19% of severe distortions occurred on slopes greater than 30°. For descending imagery, 90.47% of the distortions identified by LSM/LG, including layover and shadow, were found on slopes between 20° and 60°. The R-Index detected almost no active layover on gentle slopes below 30°, whereas the P-NG method identified 87.84% of severe distortions on slopes above 30°. All three methods consistently showed an increase in distortion proportion in steeper terrain, with the ratio of distortion proportion to slope proportion increasing significantly. Active shadow and layover tended to occur near mountain summits and ridges characterized by steep slopes, while gentler gradients more often hosted passive distortions.

The aspect facing towards or away from the satellite significantly influenced distortion occurrence. In ascending imagery, only 10.9% of distortions identified by LSM/LG were on north-, southeast-, and south-facing slopes. The R-Index detected 98.25% of active layover on southwest-, west-, and northwest-facing slopes, facing towards the satellite. The P-NG method noted the highest proportion of severe distortions, 19.3%, on north-facing slopes, indicating no clear preference for slopes perpendicular to the radar line of sight. In descending imagery, northeast-, east-, and southeast-facing slopes accounted for 77.05% of severe distortions identified by LSM/LG, 98.58% of active layover by the R-Index, and 17.34% of severe distortions by P-NG.

Compared to the LSM algorithm, the LG method effectively addressed the inability to identify passive distortions, and improved the recognition of foreshortening and resolution-enhancing areas. In the LSM method, pixels classified as valid data were further subdivided into resolution-enhancing and foreshortening categories, exhibiting higher proportions than those identified by the P-NG method.

### 3.3. Verification of Geometric-Distortion Identification Accuracy

#### 3.3.1. Verification of Geometric-Distortion Morphology in Local Areas

Figure 10 displays the distortion-identification results for a local area in the ascending orbit, analyzed using four different methods.

The LSM method shows passive shadow consistently positioned to the right of active shadow, often appearing in pairs. The sequence of layover distortion along the ground distance progresses from near-passive layover to active layover, and finally to far-passive layover. The R-Index method identifies a smaller active-layover area compared to other methods, with several regions failing to recognize layover and passive distortions. The P-NG method detects a significant number of passive shadow, sometimes exceeding the area of active shadow, with passive shadows appearing before active ones without any occlusion. Active-layover distortions also occur in isolation, scattered across various locations. The area of passive layover identified by P-NG is larger than that recognized by other methods. The severe-distortion distribution identified by the LG method is consistent with that of the LSM method, and the area of foreshortening aligns closely with the R-Index results. After categorizing less-severe distortions as either perspective shrinkage or resolution enhancing, the boundaries of various distortion types closely relate to the watershed lines and waterline of the terrain, highlighting their anastomotic relationship.

Figure 11 presents the distortion-identification results for a local area in the descending orbit, using the same four methods.

The distribution patterns of SAR geometric distortions in descending imagery align with those observed in ascending imagery, albeit with some detailed differences. In the LSM method results, passive shadow is positioned to the left of active shadow. On the northwest side of area A1, shadow distortion is surrounded by layover, a pattern also observed in the ascending results. A minimal amount of layover is present in areas A2 and A4. The R-Index results continue to show a reduced active-layover area, especially in area A4. The passive-distortion area identified by P-NG is larger than the active-distortion, particularly noticeable in the far-passive layover and passive-shadow regions. The LG method maintains LSM’s refined classification of distortions, effectively preserving layover-shadow areas and identifying smaller areas of severe distortion.

These methods reveal that the positions of shadow and layover distortions are reversed between ascending and descending images. In the ascending imagery, the eastern steep slope of area A4 displayed layover distortions, while the western slope was predominantly affected by shadow. Conversely, in descending imagery, shadow distortions occurred on the eastern slopes of A4, and the western slopes exhibited layover distortions. Despite differences in local incidence angle (LIA) and terrain details, the areas affected by layover and shadow distortions in ascending images were approximately equal, yet layover distortions significantly outweighed shadow distortions in descending images. This was particularly pronounced in area A4, characterized by north-south gullies, where shadow- and layover-distortion types were interchanged between ascending and descending images, with area A1 complementing as an eastern slope.

Moreover, severe distortions were prominent in the more complex terrains of areas A2 and A3 in both ascending and descending images. In area A1, with an ascending local incidence angle of approximately 42.5°, shadow distortions were discretely distributed at the foot of the slope, indicating subsidence away from the satellite. The satellite-facing slopes of area A2 predominantly exhibited layover distortions, which were associated with lower effective monitoring point density. Area A3, densely vegetated, resulted in sparse monitoring pixels. In area A4, effective monitoring pixels were distributed on both sides of the gully, with none in the shadow regions. In descending images, most of area A1 was covered by layover and shadow, with monitoring pixels concentrated in zones of subtle distortion. The southeastern corner of area A2 experienced interference from discrete layover and shadow, with discrete monitoring pixels on the northwestern side of the region. Area A3, despite fewer effective monitoring pixels, exhibited greater deformation values than other areas, suggesting that phase discontinuities may reduce the reliability of these measurements. Most effective monitoring values in area A4 were contained within the active layover zones, with larger deformation values at the edges of shadow distortions. Optical imagery generally confirmed that distortion edges aligned with watersheds, with higher effective values in sparsely vegetated areas than in densely vegetated ones. Terrain features and satellite flight direction also influenced the effective monitoring point density; slopes perpendicular to the radar’s LOS and facing the satellite maintained good coherence even under severe distortion.

#### 3.3.2. Accuracy Validation of Effective Monitoring Point Density

Statistical analysis (Table 6) shows that the LG method achieves superior classification across all types of geometric distortions, significantly enhancing the effective monitoring point density for subtle distortions compared to other methods. 

This article assesses the accuracy of various distortion identification methods by statistically analyzing the effective monitoring values for different types of distortions across ascending and descending orbital tracks. Specifically, after identifying distortions in 246,040,000 pixels within the study area, the study quantifies the effective monitoring pixels—166,580,460 from the ascending track and 113,659,020 from the descending track—distributed among various types of distortion pixels. Compared to three conventional methods—LSM, R-Index, and P-NG—the LG approach demonstrates distinct advantages in refining the classification of subtle distortions, such as foreshortening and resolution enhancing, and in the effective identification of severe distortions, including layover and shadow, by utilizing ground range slope. Additionally, the effective monitoring point density for areas with severe active distortions in the LG method exceeds that of passive distortion areas, unlike the P-NG method, where effective monitoring values for active shadow in ascending images are lower than those for passive shadow. 

Among the various methods, the areas of foreshortening exhibited the highest effective monitoring point density, reaching up to 62.88%, with layover areas performing better than those classified as having good visibility. Conversely, shadow areas yielded the least effective results, with a minimum of 16.49%. The LSM method’s amalgamation of subtle distortions impacts the effective monitoring point density, with ascending data showing lower values for usable areas compared to layover areas. The observed differences in effective monitoring values across different distortion types reflect the varied impact of distortions on InSAR monitoring results. The range of effective monitoring point densities identified by the LSM method among different distortions was significantly broader than those identified by other methods, highlighting more distinct characteristics among the various distortions. No significant differences were found in the density of effective monitoring points for layover distortions identified by the P-NG method; however, in ascending images, the metric for active shadow was noted to be lower than for passive ones. The R-Index demonstrated that the density for active layover was slightly less than that identified by the other three methods, with subtle distortion areas displaying values close to those found by the LG method.

Relative to the LSM method, the LG method showed an overall increase in effective monitoring point density for layover areas, while most shadow distortion monitoring values decreased, with the largest difference being 1.05% in ascending layover shadow areas. Following valid data segmentation, the density values for perspective shrinkage and resolution enhancing areas were similar to those identified by the R-Index.

## 4. Discussion

The inherent characteristics of SAR side-looking imaging critically influence the sensitivity of InSAR measurements, while the interplay between incident and terrain parameters dictates the efficacy of imaging outcomes. In mountainous regions characterized by significant terrain variations, SAR imagery produced via side-looking techniques is often marred by severe geometric distortions. These distortions significantly impair SAR’s detection capabilities and operational scopes, creating areas that effectively become observational blind spots [44]. Consequently, in practical applications, it is crucial to quantitatively pre-analyze the applicability and visibility of SAR based on DEM and SAR geometric parameters. This analysis involves the precise extraction of layover and shadow regions and the selection of appropriate SAR images to minimize the impact of these distortions. The extent of geometric distortions can be effectively delineated, and while areas of foreshortening and resolution enhancement can be mitigated through geometric transformations, layover and shadow regions typically require estimation or masking during processing to minimize their impact on phase unwrapping [12].

Two principal strategies emerge for the effective identification and extraction of geometric distortions. The first relies on the statistical properties of SAR signals, utilizing assumptions such as Gaussian distribution in layover areas to facilitate their identification [45], or employing methods that combine kernel density functions with a constant false alarm rate to delineate geometric distortions [46]. The second strategy hinges directly on the relative geometric relationship between the sensor and the Earth’s surface, which allows for the accurate determination of distortion extents, including the use of terrain shadow models for distortion classification [47] that assume a constant local incidence angle. While statistically based methods can yield certain outcomes, they are susceptible to threshold settings; hence, geometry-based methods offer greater accuracy and reliability.

This study utilized effective monitoring point density as a benchmark to evaluate the recognition accuracy of four SAR geometric distortion identification methods—LSM, R-Index, P-NG, and LG—across complex terrains. By analyzing the coupling relationship between the distribution of identified geometric distortions and topographical changes, the results indicate that the LG method, while retaining high recognition accuracy, comprehensively classified all pixel-level geometric distortions within the study area. This approach provides a novel way to mitigate the impact of geometric distortions and ensures the reliability of InSAR deformation monitoring data. These findings hold significant implications for the application of SAR data in deformation monitoring and contribute to enhancing the accuracy of deformation classification in complex terrains.

Along the radar LOS, the LSM method achieved the most logical distribution order for near-passive layover, active layover, far-passive layover, active shadow, and passive shadow, without any overlap among distortion types. By integrating effective monitoring-point density with optical satellite imagery, the LSM method’s classification of distortions aligned most consistently with SBAS monitoring results. Conversely, the P-NG method identified severe distortions predominantly on slopes perpendicular to the satellite’s flight direction due to its non-adjustment for ground range slope. Although the P-NG method significantly differentiated between shadow and layover distortions in terms of effective monitoring-point density, it lacked clarity in distinguishing between active- and passive-distortion types. The R-Index, which represents the ratio between pixel sizes in the imaging plane and on the ground—the “pixel compression factor”—identified active layover aligning well with the active-distortion areas detected by LSM. The R-Index did not explicitly categorize active shadow; however, conditions implying $90{}^{\circ}-\theta_{ij}>S_{ij}$ combined with Formula (2) suggested that active shadows were likely considered as good visibility areas. The LG method inherited LSM’s precise distortion-classification traits and excelled in accurately categorizing subtle distortions.

Within the ascending-orbit images, the discrepancy in the number of active-layover pixels identified by the LSM method compared to those identified using principles of geometric distortion was 7.53 per thousand, with a disparity of 3.21% for active-shadow pixels. In descending-orbit images, the discrepancy for active-layover pixels was 2.48‰; for active-shadow pixels, it was 5.04%. Across the entire study area, the most substantial proportion of active layover, at 7.32%, was observed in the descending orbit. Given these findings, the disparities between the two identification methods are minimal, facilitating the integration of both methods to form the LG method that balances accuracy and comprehensiveness.

The P-NG and the other three methods (LSM, R-Index, and LG) exhibit apparent differences in how slope is incorporated, with the latter group utilizing ground-range slope, while P-NG directly compares slope with LIA. The distinction between using raw slope and calibrated ground-range slope in identifying the four classes of distortions—foreshortening, active layover, active shadow, and resolution enhancing—in descending data is depicted in Figure 12. The results obtained using ground-range slope are consistent with those from the R-Index and LG approaches, efficiently minimizing misidentified severe distortions. Quantitatively, as shown in Table 7, the proportion of active layover decreased from 10.16% to 2.48%, and active shadow was reduced from 1.24% to 0.30%. As anticipated, the classification of mountainous distortions parallel to the radar LOS significantly reduced severe distortions compared to classifications using raw-slope values.

Notably, the DEM used in this study has a resolution of 5 m, which may not be well-suited for very high-resolution SAR data. Future research could explore using higher resolution DEMs or employing deep learning techniques to improve geometric distortion identification in extremely high-resolution SAR data. Employing the results obtained in this study as training samples for deep learning models aimed at identifying SAR geometric distortions offers a promising methodology for achieving accurate and reliable identification results. Future research may focus on extending this approach to handle high-resolution SAR data and explore further enhancements by utilizing deep learning technologies.

## 5. Conclusions

This study utilized LSM, R-Index, P-NG, and LG methods to extract geometric distortions within Sentinel-1 ascending and descending SAR imagery of the research area. Surface-deformation monitoring results for the region were derived from multi-temporal ascending- and descending-track SAR images in conjunction with SBAS-InSAR. By incorporating local topography, a large-scale comparison of the precision of these four distortion-identification methods was conducted. The findings are as follows:(1)Terrain characterized by steep slopes greater than 30 degrees is more susceptible to SAR geometric deformation. For such topographies, employing SAR imagery with smaller LIA enhances the accuracy of deformation monitoring.(2)The terrain distortions identified by the LSM, LG methods, and R-Index strongly correlate with slope orientation and the LOS direction. Pre-calculating the ground-range slope before conducting SAR geometric-deformation analysis can significantly improve the accuracy of geometric-distortion identification.(3)Among traditional methods, the LSM is best suited for regions with complex mountain ridges and rugged terrain featuring significant slopes, where there is a lesser demand for subtle distortion classification. The R-Index effectively describes the compression of individual pixels along the range direction. The P-NG method offers advantages in areas dominated by north-south-oriented mountain ranges.(4)The LG method identifies in detail all types of geometric distortions. It retains the high identification precision of the LSM method and further subdivides subtle distortions into zones of resolution enhancing and foreshortening, resulting in clearer monitoring outcomes. The substantial differences in effective monitoring-point density for regions with larger subtle distortion areas after this partitioning underscore the necessity of this division.

In practical InSAR monitoring, pre-identifying distortions in the monitoring area enables the selection of appropriate SAR data, thereby reducing the cost of trial and error in data acquisition. As SAR data acquisition platforms evolve, the increasing resolution of the data exacerbates the effects of signal overlap in layover areas. After identifying severely distorted regions, the utilization of masking techniques emerges as a cost-effective processing strategy, particularly in high-altitude mountainous areas. While physical models for SAR geometric-distortion recognition continue to refine, the specific impacts of various types of geometric distortions on echo signals remain to be fully understood. This paper’s precise identification of distortions lays the groundwork for refining our understanding of how different distortions affect SAR data processing.

## Figures and Tables

**Figure 1 sensors-24-02834-f001:**
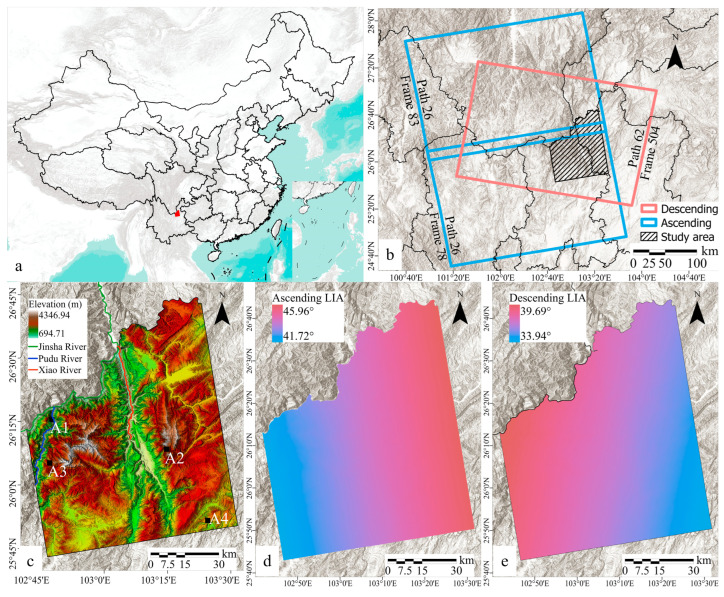
Geographical information of the study area. (**a**) Position of the research area (**b**) The location of Sentinel-1A dataset coverage (blue and pink box) (**c**) DEM of the study area and distribution of four representative local areas. (**d**,**e**) Local incident angles of ascending and descending orbits of SAR imagery within the research area.

**Figure 2 sensors-24-02834-f002:**
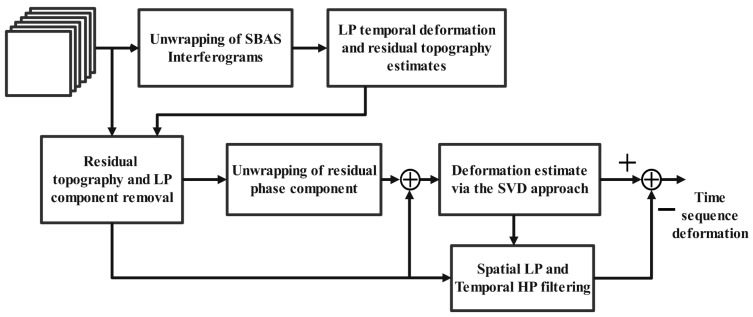
Block diagram of SBAS-InSAR [38]. Through multi-view processing of time series data, SBAS-InSAR can improve the time resolution of deformation calculations and achieve more precise monitoring of short-term changes in surface deformation.

**Figure 3 sensors-24-02834-f003:**
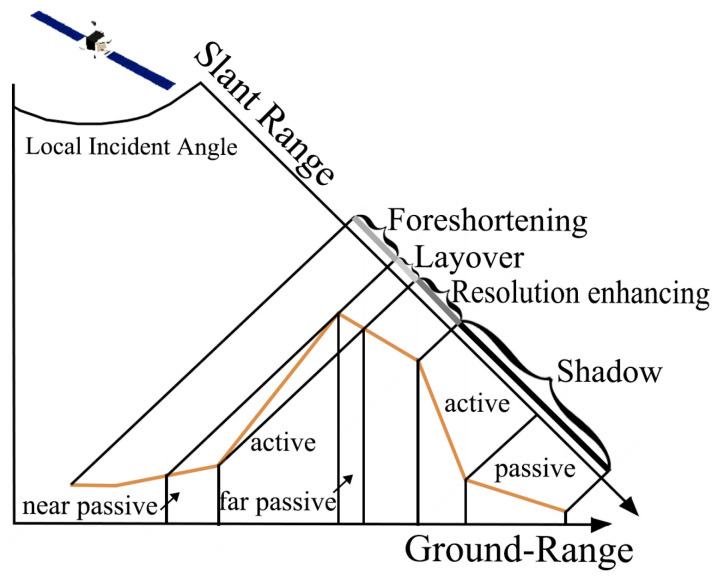
Four types of distortion determined by the relationship between the local incidence angle and terrain slope. Yellow lines represent the abstract undulating surface along the distance. The identification of passive distortion necessitates the inclusion of additional parameters.

**Figure 4 sensors-24-02834-f004:**
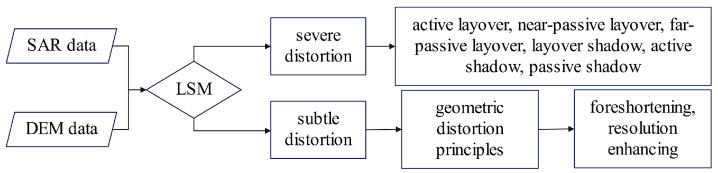
The processing chain of the proposed method.

**Figure 5 sensors-24-02834-f005:**
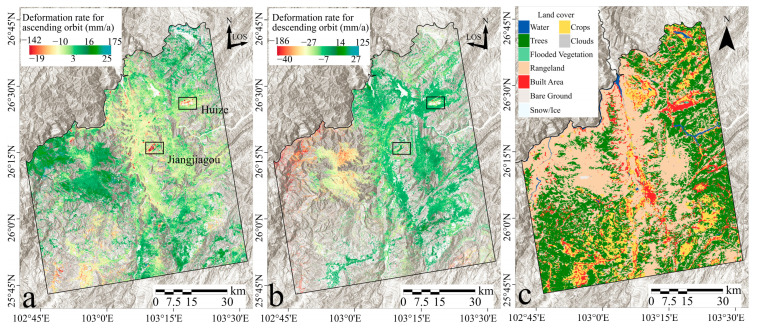
SBAS deformation monitoring results. Panels (**a**,**b**), respectively, display the deformation rates for ascending and descending orbits, with areas outside ±3σ compressed to the edges of the color bar. In panel (**a**), areas of significant deformation are highlighted and labeled with place names. In panel (**c**), the land use map is depicted, and regions where vegetation height is less than 15 m are classified as farmland and grassland.

**Figure 6 sensors-24-02834-f006:**
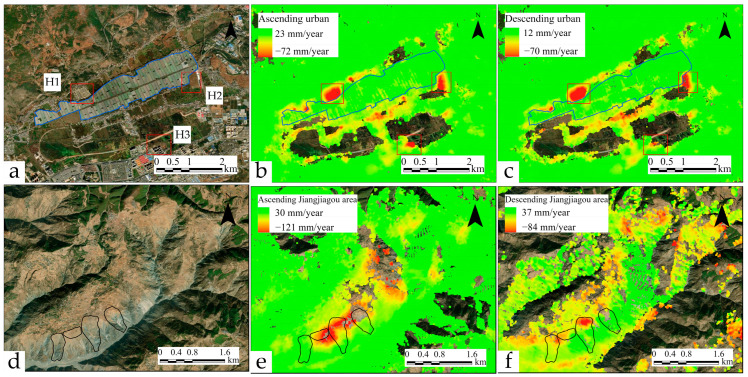
Localized deformation in the Huize urban district and Jiangjiagou. Panels (**a**–**c**) display deformation in the Huize County urban area, with notable regions annotated; blue areas highlight plastic greenhouses used for vegetable cultivation. Panels (**d**–**f**) depict deformation in the Jiangjiagou region, where black areas represent historical landslides.

**Figure 7 sensors-24-02834-f007:**
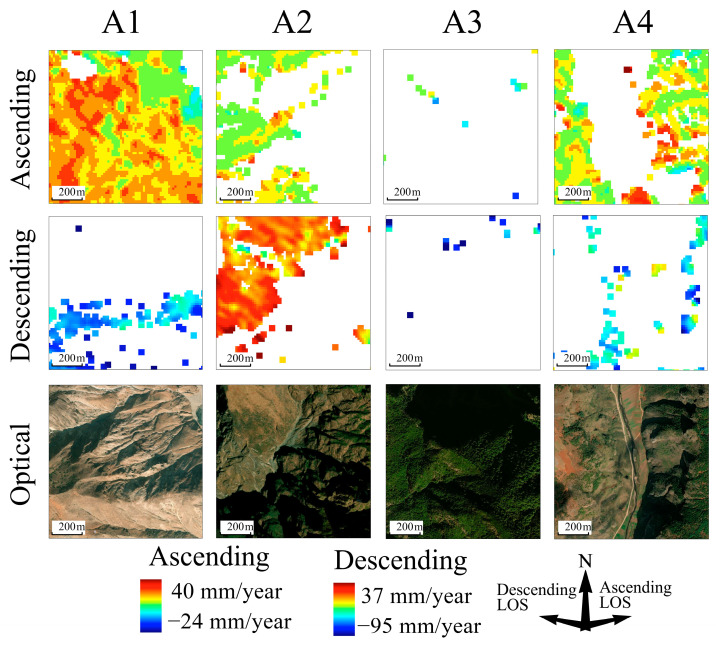
Deformation rate maps for ascending and descending orbits in four representative regions, accompanied by optical satellite images.

**Figure 8 sensors-24-02834-f008:**
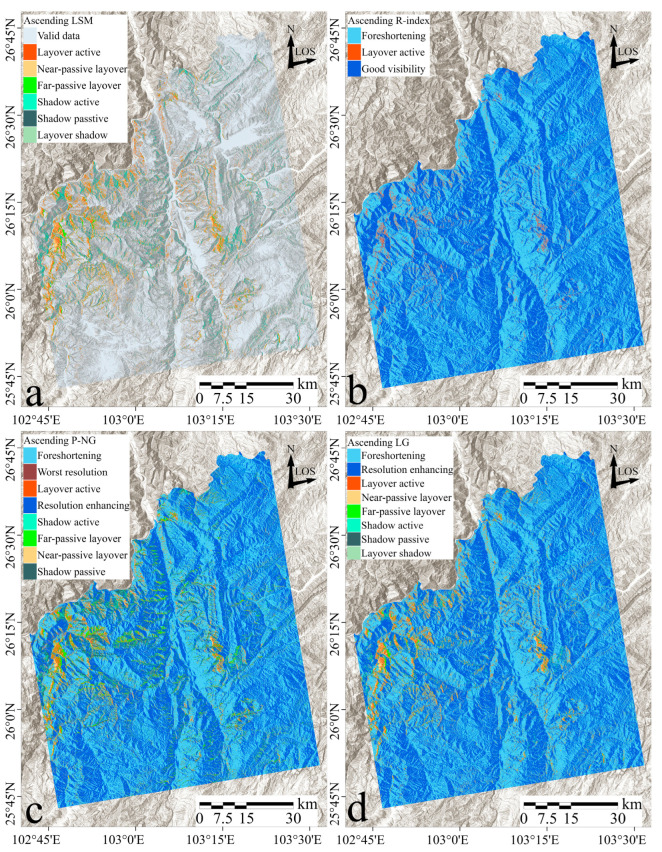
Identification results of geometric distortions in the study area using ascending-track imagery. Distortions identified by LSM, R-index, P-NG, and LG methods are displayed in panels as (**a**–**d**), respectively.

**Figure 9 sensors-24-02834-f009:**
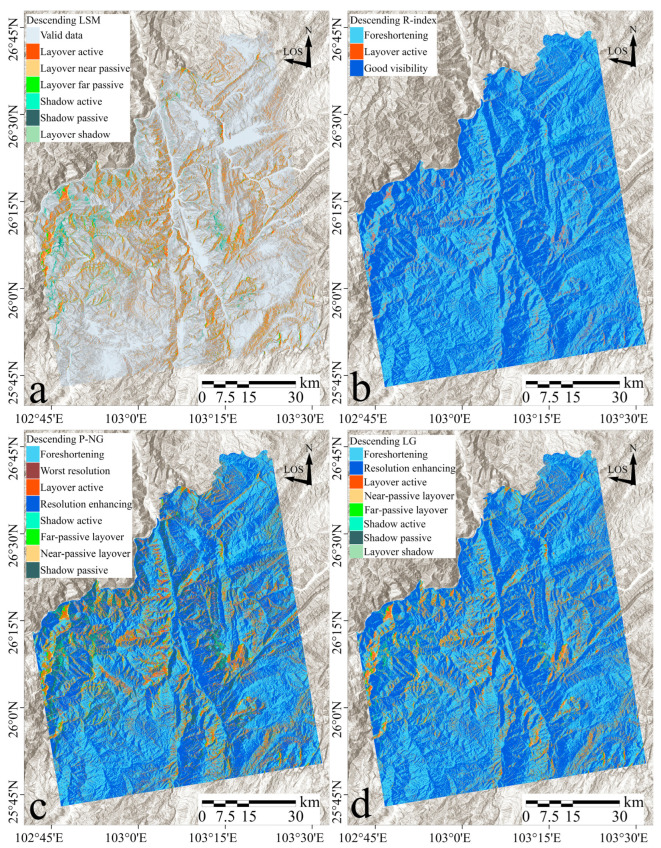
Identification results of geometric distortions in the descending track imagery. Categories identified by LSM, R-index, P-NG, and LG methods are displayed in panels (**a**–**d**), respectively.

**Figure 10 sensors-24-02834-f010:**
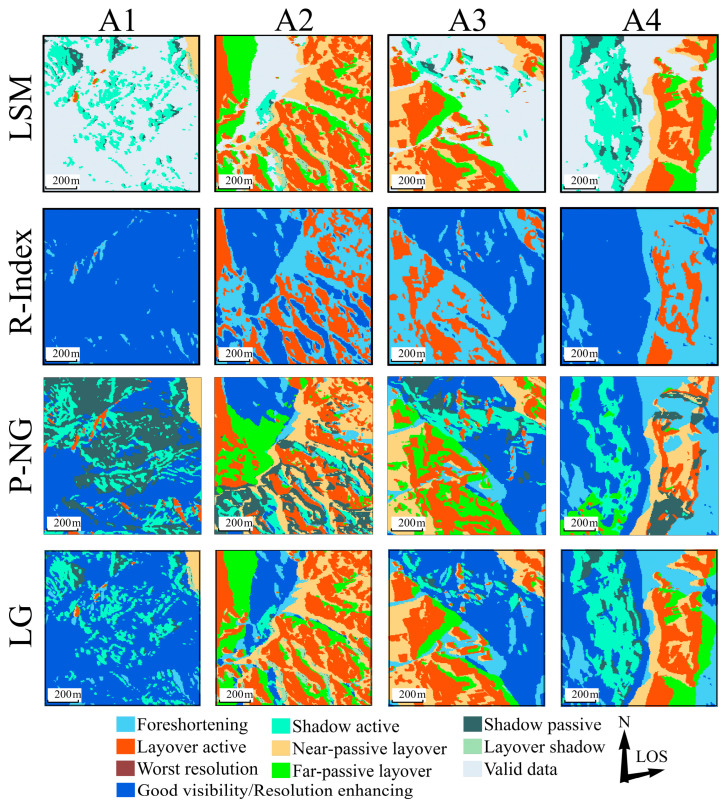
Ascending-distortion identification results of the local area, with the angle between the line of sight (LOS) and the north direction being 78°.

**Figure 11 sensors-24-02834-f011:**
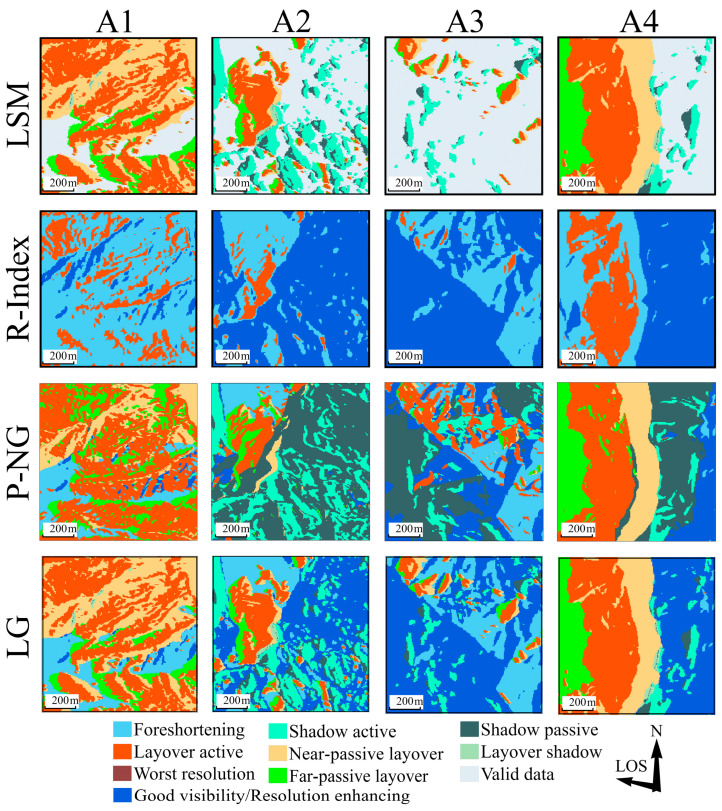
Descending distortion-identification results for the local area by the four methods, with the angle between the line of sight (LOS) and the north direction being −78°.

**Figure 12 sensors-24-02834-f012:**
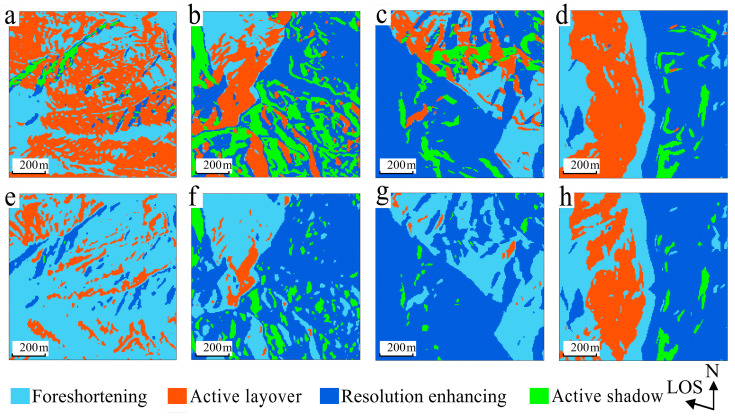
Geometric-distortion classification maps identified using distortion-recognition principles. Panels (**a**–**d**) utilize slope, while (**e**–**h**) represent ground-range slope, showing significant changes in mountainous distortion types parallel to LOS.

**Table 1 sensors-24-02834-t001:** Key parameters of Sentinel-1 imagery in the study area.

Flight Direction	Ascending	Descending
Path	26	62
Frame	83, 78	504
Radar Azimuth	347.47°	192.53°
Acquisition Time	15 January 2021–24 December 2022	5 January 2021–26 December 2022
Images	60 × 2	56

**Table 2 sensors-24-02834-t002:** Classification of distortion types.

Aspect Type	$\mathrm{LIA}\;\theta_{IJ}$ and Ground-Range Slope $X_{ij}$ (°)	Distortion Type
Facing satellite (front slope)	$\theta_{ij}\geq X_{ij}$	Foreshortening
	$\theta_{ij}<X_{ij}$	Active layover
Facing-away satellite (back slope)	$90^{\circ }-\theta_{ij}\leq X_{ij}$	Active shadow
	$90^{\circ }-\theta_{ij}>X_{ij}$	Resolution enhancing

**Table 3 sensors-24-02834-t003:** Proportion of results from different geometric-distortion-analysis methods.

Distortion Type	LSM		P-NG		R-Index		LG	
	Ascending	Descending	Ascending	Descending	Ascending	Descending	Ascending	Descending
Layover, active	3.36%	7.32%	3%	9.89%	1.99%	3.04%	3.36%	7.32%
Layover, near-passive	1.77%	3.16%	3.3%	3.56%	0	0	1.77%	3.16%
Layover, far-passive	1.11%	2.27%	1.74%	2.11%	0	0	1.11%	2.27%
Shadow, active	2.15%	1.00%	2.1%	1.15%	0	0	2.15%	1.00%
Shadow, passive	0.30%	0.28%	4.04%	8.65%	0	0	0.30%	0.28%
Layover shadow	0.11%	0.07%	0	0	0	0	0.11%	0.07%
Foreshortening	0	0	43.04%	33.64%	49.1%	45.96%	45.29%	37.05%
Good visibility	0	0	42.78%	40.98%	48.91%	51%	45.91%	48.85%
Valid data	91.2%	85.9%	0	0	0	0	0	0

**Table 4 sensors-24-02834-t004:** Statistics of severe-distortion identification results on terrains with different slopes.

Slope (°)	Percent	LSM/LG		R-Index		P-NG	
		Ascending	Descending	Ascending	Descending	Ascending	Descending
0–10	14.33%	0.71%	1.36%	0	14.33%	0.71%	1.36%
10–20	20.77%	1.99%	3.89%	0.01%	20.77%	1.99%	3.89%
20–30	25.00%	6.28%	14.01%	0.07%	25.00%	6.28%	14.01%
30–40	25.67%	26.05%	40.08%	3.17%	25.67%	26.05%	40.08%
40–50	11.12%	41.54%	26.59%	52.70%	11.12%	41.54%	26.59%
50–60	2.36%	16.20%	9.73%	28.87%	2.36%	16.20%	9.73%
60–70	0.60%	5.57%	3.34%	11.35%	0.60%	5.57%	3.34%
70–80	0.15%	1.55%	0.94%	3.52%	0.15%	1.55%	0.94%
80–90	0.01%	0.11%	0.07%	0.30%	0.01%	0.11%	0.07%

**Table 5 sensors-24-02834-t005:** Statistics of severe geometric-distortion identification results on different aspects (ascending-orbit heading 347.47°, descending-orbit heading 192.53°).

Aspect (°)	LSM/LG		R-Index		P-NG	
	Ascending	Descending	Ascending	Descending	Ascending	Descending
Flat	0	0	0	0	0	0
North	4.41%	3.70%	0	0	19.30%	7.03%
Northeast	10.76%	15.97%	0	9.00%	8.50%	13.10%
East	13.94%	33.85%	0	52.99%	5.85%	17.34%
Southeast	4.24%	27.22%	1.69%	36.58%	9.53%	16.09%
South	4.67%	7.02%	34.14%	1.42%	16.61%	11.31%
Southwest	20.28%	2.98%	53.00%	0	12.50%	5.90%
West	28.86%	5.09%	11.11%	0	13.83%	15.72%
Northwest	12.84%	4.17%	0.05%	0	13.89%	13.51%

**Table 6 sensors-24-02834-t006:** Effective monitoring point density for distortion types identified by different methods.

Distortion Type	LSM		P-NG		R-index		LG	
	Ascending	Descending	Ascending	Descending	Ascending	Descending	Ascending	Descending
Layover active	57.22%	29.66%	58.75%	32.96%	54.33%	28.53%	57.66%	29.80%
Layover near passive	47.43%	25.50%	58.19%	30.46%	0	0	47.93%	25.58%
Layover far passive	53.90%	31.71%	51.41%	25.15%	0	0	54.16%	31.80%
Shadow active	36.01%	27.73%	39.14%	30.95%	0	0	35.97%	27.69%
Shadow passive	32.28%	16.54%	45.73%	25.03%	0	0	32.40%	16.49%
Layover shadow	51.04%	30.36%	0	0	0	0	52.09%	30.63%
Worst resolution	0	0	60.00%	80.00%	0	0	0	0
Foreshortening	0	0	60.89%	39.20%	62.05%	38.90%	62.88%	41.09%
Good visibility	0	0	39.51%	32.83%	41.96%	32.92%	41.98%	33.26%
Valid data	52.39%	36.65%	0	0	0	0	0	0

**Table 7 sensors-24-02834-t007:** Distortions classified using slope or ground-range slope.

Distortion Type	Slope and LIA	Ground Range Slope and LIA
Foreshortening	38.82%	46.50%
Active layover	10.16%	2.48%
Resolution enhancing	49.78%	50.72%
Active shadow	1.24%	0.30%

## Data Availability

Data are contained within the article.

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
