# Peer review of "Accuracy Assessment of Geometric-Distortion Identification Methods for Sentinel-1 Synthetic Aperture Radar Imagery in Highland Mountainous Regions"

_sensors, 2024, doi:10.3390/s24092834_

Round 1

Reviewer 1 Report

Comments and Suggestions for Authors

Reviewer 2 Report

Comments and Suggestions for Authors

This manuscript (Sensors-2949912) introduces an innovative LSM-ground range slope (LG) method that significantly improves the identification and classification of subtle geometric distortions in SAR imagery. It is particularly effective in areas with complex terrains, such as the Xiaojiang, Pudu, and Jinsha River regions. The LG method is demonstrated to be superior to traditional approaches, providing detailed discrimination and comprehensive recognition of distortions. This approach enhances the accuracy of environmental and geological monitoring.

However, in its current form, the manuscript is not suitable for publication.

Regarding the repetition of keywords with the paper's title; what is the purpose of section “2. Study Area and Data 98”, and why is not it integrated with the materials and methods section? All figure and table captions require rewriting to accurately and precisely describe all displayed elements. In Figure 1, why obscure the map with an overlay in pink? This needs to be revised. What do the negative values indicate in Figure 5? For instance, the flowcharts are not adequately explained, with the figures merely placed in the text. In this same materials and methods section, few references are cited, and an overly descriptive and confusing narrative is provided. The results section is dense and confusing. The images are not adequately described, leading to a mix-up of discussion and the authors' opinions. There is a blurring of lines between materials, methods, and results. Figures and tables are underutilized, making it challenging to understand the authors' intentions, often without proper reference to the figures. Statistical analyses are lacking, diminishing the emphasis on the explored subject matter. The discussion section is also inadequate. The authors have not fully explored what was achieved and the advancements found in relevant literature. Moreover, the discussion is disproportionately brief compared to other sections of the manuscript. The conclusion is presented in bullet points, summarizing the manuscript's content without providing appropriate perspectives. References need to be updated. The language requires revision for clarity, and many sections and lengthy paragraphs make reading and understanding challenging.

Comments on the Quality of English Language

English grammar and spelling also require significant improvements.

Round 2

Reviewer 2 Report

Comments and Suggestions for Authors

The manuscript (sensors-2949912) has significantly improved. The sections, captions, and figures are now appropriate. The language has also significantly improved. But small adjustments are still needed. The authors should combine the two sections “2. Study Area and Data” and “3. Methods”, as the study area and the subsections are part of what we call “Material and Methods”. Another question to be answered in the manuscript. How many simulations were performed to calculate the significant values?

Comments on the Quality of English Language

Minor changes in grammar and spelling.

Author Response

Thank you very much for your constructive comments and insights regarding our manuscript. Your guidance has been invaluable to the advancement of our research.

In response to your suggestions, we have made the following revisions and enhancements to the manuscript:

1. Section Merging: We have merged the sections "2. Study Area and Data" and "3. Methods" into a single section titled "2. Materials and Methods". This adjustment provides a unified and clearer presentation of the materials and methods involved, enhancing the overall structure of our research.

2. Language Polishing: We have conducted another thorough revision of the manuscript to enhance the clarity and precision of the language, ensuring all potential grammatical errors have been addressed.

3. Significant Values: To address your query regarding the number of simulations performed for calculating significant values, we have added detailed descriptions between lines 535 and 540 of the manuscript. Specifically, "This article assesses the accuracy of various distortion identification methods by statistically analyzing the effective monitoring values for different types of distortions across ascending and descending orbital tracks. Specifically, after identifying distortions in 246,040,000 pixels within the study area, the study quantifies the effective monitoring pixels—166,580,460 from the ascending track and 113,659,020 from the descending track—distributed among various types of distortion pixels."

We hope these revisions meet your expectations and further enhance the quality of the manuscript. We appreciate your meticulous review and valuable feedback, and look forward to any further suggestions you might have.

Thank you once again for your attention and assistance.